# Effect of Ultrasound-Assisted Sodium Bicarbonate Treatment on Aggregation and Conformation of Reduced-Salt Pork Myofibrillar Protein

**DOI:** 10.3390/molecules27217493

**Published:** 2022-11-03

**Authors:** Zhuang-Li Kang, Xue-Yan Shang, Yan-Ping Li, Han-Jun Ma

**Affiliations:** 1School of Tourism and Cuisine, Yangzhou University, Yangzhou 225127, China; 2School of Food Science, Henan Institute of Science and Technology, Xinxiang 453003, China

**Keywords:** ultrasound, sodium bicarbonate, myofibrillar protein, hydrophobicity, Zeta potential

## Abstract

To study the effects of an ultrasound (0, 30, and 60 min) and sodium bicarbonate (0% and 0.2%) combination on the reduced-salt pork myofibrillar protein, the changes in pH, turbidity, aggregation, and conformation were investigated. After the ultrasound-assisted sodium bicarbonate treatment, the pH increased by 0.80 units, the absolute value of Zeta potential, hydrophobic force, and active sulfhydryl group significantly increased (*p* < 0.05), and the turbidity and particle size significantly decreased (*p* < 0.05). Meanwhile, the fluorescence intensity decreased from 894 to 623, and the fluorescence peak showed a significant redshift, which indicated that the ultrasound-assisted sodium bicarbonate treatment exposed the non-polarity of the microenvironment in which the fluorescence emission group was located, leading to the microenvironment and protein structure of myofibrillar tryptophan being changed. Overall, an ultrasound-assisted sodium bicarbonate treatment could significantly improve pork myofibrillar protein solubility and change the protein structure under a reduced-salt environment.

## 1. Introduction

In recent years, consumers have paid more and more attention to healthy emulsified meat products. One of the main functions of salt in emulsified meat is the dissolution of myofibrillar proteins [1,2]. Salt promotes the dissolution and swelling of myofibrillar protein, extracts more myofibrillar proteins from the meat, and endows meat products with good taste and flavor [3,4,5]. Excessive dietary salt intake can cause cardiovascular disease and other diseases [6]. A previous study reported that meat products account for approximately 20% of people’s normal daily unrestricted sodium intake and is considered to be the second-largest contributor of sodium in their diet [7]. Therefore, research into how to reduce the quantity of salt in meat products is a challenge for research [8,9].

Sodium bicarbonate, a generally-recognized-as-safe (GRAS) food ingredient, contains a synergistic anion (HCO_3_^–^), a stronger buffer capacity, and is widely used in meat products [10]. Myofibrillar protein is the main component of protein in the process of gel network formation and is mainly responsible for the structure and functional characteristics of composite gel meat products’ key protein. Meanwhile, myofibrillar protein is an important component of pork batter, accounting for approximately 55% of its protein. It not only gives meat products flavor, taste, and nutrition, but also has multiple functional characteristics such as water-holding, emulsifying, and gel properties [11]. Our previous study found that sodium bicarbonate replacing a certain amount of salt can increase the solubility of the myofibrillar protein of pork and lower the protein aggregation, improving the water-holding capacity and textural properties of pork myofibrillar protein gel [5]. However, there is a lack of information as to the approximate application of ultrasound-assisted sodium bicarbonate treatment in emulsified meat products.

Ultrasound technology has been widely used in food processing because of its high efficiency, low cost, and high level of safety [12,13,14,15]. The cavitation and mechanical action of ultrasound treatment can destroy the force between proteins, make their structure loose, and accelerate protein dissolution [16,17]. The cavitation effect of ultrasound technology can significantly affect the conformation and structure of proteins due to it resulting in an improvement of solubilization, viscosity, gelation, flavor binding, and interfacial properties of proteins [5,18]. Zhang, Regenstein, Zhou, and Yang [19] reported that ultrasound treatment (600 W) has a denser and more uniform gel microstructure, which improved the water-holding capacity (WHC) of the gels. Moreover, high-power ultrasound (20 min, 300 W) technology improved the gel properties and thereby allowed for a partial reduction in the salt levels in chicken meat batters [11]. A previous study found that the ultrasound-assisted sodium bicarbonate promoted actomyosin denaturation of a chicken breast, increased the α-helix content and cooking yield, and reduced the fluorescence intensity of tyrosine and tryptophan [20]. Our previous study reported that ultrasound-assisted sodium bicarbonate treatment can improve gel characteristics and the water-holding capacity of reduced-salt pork batters [21]. However, to our knowledge, few studies exist on the mechanism of ultrasound-assisted sodium bicarbonate on the molecular level of pork myofibrillar protein. Therefore, the aim of this study was to explore the effect of ultrasound-assisted (0, 30, and 60 min) sodium bicarbonate (0% and 0.2%) on the solubility, surface hydrophobicity, active sulfhydryl group, and fluorescence of reduced-salt pork myofibrillar protein, and to determine the influencing mechanism of ultrasound-assisted sodium bicarbonate.

## 2. Results and Discussion

### 2.1. pH

The effects of the ultrasound-assisted sodium bicarbonate treatment on the pH of myofibrillar protein solution is shown in Figure 1. The pH increased significantly (*p* < 0.05) after the addition of sodium bicarbonate. Sodium bicarbonate is well-known to be a strong base weak acid salt; it can produce OH^−^ and HCO_3_^−^ in the water, leading to a pH increase. A previous study reported that the pH of myofibrillar protein solution increases significantly with the addition of sodium bicarbonate from 0% to 0.42% [22]. Moreover, the pH increased significantly (*p* < 0.05) with the increase in ultrasonic treatment time because the ultrasound treatment increases the local temperature and pressure in the area around the bubbles generated by the ultrasound wave. This leads to protein denaturation, the generation of free radicals that react with protein side-chains, the reduction in acidic protein groups, and the rise in the pH level [16]. Li, Kang, Zhao, Xu, and Zhou [11] reported that ultrasound treatment for 6 min causes the pH of PSE batter to increase. A similar study has shown that the pH of raw pork batter increased significantly with the increase in ultrasound time and sodium bicarbonate [21].

### 2.2. Turbidity

Turbidity is an indicator reflecting the degree of protein aggregation. The decrease in turbidity indicates the dispersion of the myofibrillar protein complex, while the increase in turbidity indicates the aggregation of the complex [23]. The effect of ultrasound-assisted sodium bicarbonate on the turbidity of pork myofibrillar protein is shown in Figure 2. With the increase in ultrasound time and the addition of sodium bicarbonate, the turbidity decreased significantly (*p* < 0.05). The main reason for this change is that the pH increased significantly (Figure 1), which led to the pork myofibrillar protein shifting away from the isoelectric point and promoting the dissolution of the protein [5,24]. The decreased turbidity was also due to strong physical forces such as shear force, shock waves, and turbulence generated by the ultrasonic treatment, which break down the protein particles in the solution system, reduce the turbidity, and make the solution clear [25].

### 2.3. Particle Size and Zeta Potential

The particle size is one of the factors that affects the functional properties of the protein and the macroscopic expression of protein structure [26]. Figure 3A shows that the particle size of the pork myofibrillar protein solution significantly decreased (*p* < 0.05) with an increase in the ultrasonic treatment time and the addition of sodium bicarbonate. The result was in agreement with the result of turbidity (Figure 2). Some studies found that more myofibrillar protein could be dissociated to form smaller molecules with the increase in sodium bicarbonate, which increases the pH of the myofibrillar protein [22,27]. Another factor is the strong physical force generated by ultrasonic cavitation, wherein the protein particles can be effectively broken and the particle size reduced [28]. Arzeni et al. [29] reported that ultrasound can reduce the particle size of soybean protein particles, and make the particle size of the solution more uniform.

In contrast, the Zeta potential can reflect the surface charging property of particles in the solution, and describe the electrostatic interaction between particles [30]. Generally speaking, when the number of positively charged amino acids in the protein solution system is higher than that of negatively charged amino acids, the potential of the protein solution is positive [31]. As shown in Figure 3B, the Zeta potential is negative, indicating that the number of negatively charged amino acids on the surface of the myofibrillar protein is higher than that of the positively charged ones. With the increase in ultrasound time and the addition of sodium bicarbonate, the absolute value of Zeta potential in the myofibrillar protein increased significantly (*p* < 0.05). The reason for this change may be that after the ultrasound-assisted sodium bicarbonate treatment, the molecules unfolded and more negatively charged amino acids were exposed, resulting in an increase in the net negative charge and aggregation of protein molecules being destroyed. Meanwhile, with the increase in pH, there is an ionization of the amino acids free-carboxyl groups on the surface, resulting in the Zeta potential becoming more negative. Some studies reported that ultrasound can cause the unfolding of some protein structures, and then increase the binding sites of protein surface charges, leading to an increase in the absolute value of Zeta potential. The result therefore helps to enhance the repulsion force between proteins, thus hindering protein aggregation [19,32]. In addition, the greater the absolute value of Zeta potential, the stronger the interaction between protein and water, and the better the degree of protein dissolution [33].

### 2.4. Active Sulfhydryl

The sulfhydryl group is one of the most active functional groups in protein. It plays an important role in stabilizing and maintaining the spatial structure of proteins, and is an important factor affecting the functional characteristics of proteins [34]. Active sulfhydryl refers to the sulfhydryl group exposed to the surface of protein molecules. The effects of ultrasound-assisted sodium bicarbonate treatment on the active sulfhydryl of raw pork myofibrillar protein are shown in Figure 4. With the increase in ultrasound time and the addition of sodium bicarbonate, the absolute value of Zeta potential in the myofibrillar protein significantly increased (*p* < 0.05) from 18.04 to 25.6. The main reason is that the partial hydrolysis of myofibrillar protein molecules is promoted by sodium bicarbonate, which caused the internal sulfhydryl to be exposed to the molecular surface [22]. Moreover, ultrasound could destroy the structure of the myofibrillar protein with the increase in ultrasonic treatment time. Meanwhile, the protein molecules gradually unfold, so that the active sulfhydryl content reaches the maximum after ultrasound-assisted sodium bicarbonate treatment, and then promotes the formation of disulfide bonds within the protein molecules due to the increase in the active sulfhydryl content [29]. Some researchers have found that the ultrasonic treatment increased the reactive sulfhydryl group of myofibrillar protein from 3.67 to 7.33, indicating that the sulfhydryl groups inside the protein are brought to the surface and increase the active sulfhydryl groups [19]. A similar study has been reported by Wang, Yang, Tang, Ni, and Zhou [35], who found that the reactive sulfhydryl groups of chicken myofibrillar protein increased significantly within a longer pulsed ultrasonic treatment time (3–6 min). Thus, ultrasound-assisted sodium bicarbonate treatment promoted the stretching and unfolding of myofibrillar protein molecules, and the internal sulfhydryl groups were brought to the surface to increase the active sulfhydryl content significantly.

### 2.5. Surface Hydrophobicity

The surface hydrophobicity can directly reflect the exposure of hydrophobic amino acids on the protein surface and the degree of protein denaturation [36,37]. As shown in Figure 5, the surface hydrophobicity of myofibrillar protein significantly increased (*p* < 0.05) from 62.46 μg to 73.59 μg with the increase in ultrasound time and the addition of sodium bicarbonate. According to the study conducted by Li, Zhang, Lu, and Kang [22], the addition of an appropriate amount of sodium bicarbonate can reduce the curl of the internal protein hydrophobic groups, enhancing the surface hydrophobicity of the protein. Because the addition of sodium bicarbonate could make the pH deviate from the isoelectric point, leading to the movement of hydrophobic amino acid residues in the protein, thus increasing the repulsion between protein molecules and increasing the hydrophobic group [38]. For the other sample, an ultrasound treatment for 60 min increased BPB bound by approximately 5 μg compared with the sample without ultrasound. This may be because ultrasound produces a hole effect and microfluidic beam, which can expose many hydrophobic groups embedded in the protein molecules or polymers; thus, increasing the surface hydrophobicity of the myofibrillar protein solution [39]. The result was the opposite of the result of turbidity (Figure 1). However, Walayat et al. [40] found a logical decrease in solubility in myofibrillar proteins when surface hydrophobicity increased. The main reason for the variance in results is the different processing methods and added substances. Moreover, the hydrogen bond between protein molecules and water may break and new hydrophobic groups may appear under the action of ultrasound [37]. A previous study showed that ultrasound treatment can increase the surface hydrophobicity of a soy protein isolate solution [39]. Thus, the use of an ultrasound-assisted sodium bicarbonate treatment can increase the surface hydrophobicity of the myofibrillar protein.

### 2.6. Fluorescence Emission Spectra

The endogenous fluorescence of proteins mainly comes from aromatic amino acid residues of proteins, such as tryptophan, tyrosine, and phenylalanine [41]. The changes can reflect the changes in protein structure. The endogenous fluorescence mainly analyzes tryptophan residues, which can eliminate the interference of tyrosine residues at 295 nm, and can be used to monitor the change in protein structure [42]. As shown in Figure 6, the maximum emission wavelength of tryptophan is approximately 336 nm. With the increase in ultrasonic treatment time and the addition of sodium bicarbonate, the fluorescence intensity and maximum emission wavelength were significantly affected. The fluorescence peak product occurred redshift, and the fluorescence intensity of the sample treated with ultrasound for 60 min reached a minimum value. The redshift of the fluorescence peak indicated that the nonpolarity of the microenvironment in which the fluorescence emission group is located increases, exposing more residue of the tryptophan to the hydrophilic environment [43]. When the myofibrillary protein was treated with ultrasound-assisted sodium bicarbonate, the maximum emission wavelength shifted by 1.5 nm, and the fluorescence intensity decreased significantly (*p* < 0.05). This may be due to the ultrasound-assisted sodium bicarbonate treatment unfolding the protein structure, which causes tryptophan to be exposed to the external polar environment. The polar solvents can then cause fluorescence quenching of chromophores and reduce endogenous fluorescence intensity [44]. Thus, an ultrasound-assisted sodium bicarbonate treatment could influence the tryptophan microenvironment and change the protein structure of the pork myofibrillar protein.

## 3. Materials and Methods

### 3.1. Raw Materials and Ingredients

Chilled pork (*Duroc* × *(Landrace* × *Yorkshire)*, (180 ± 3 days old, 100 ± 5 kg)) lean leg meat (*mesoglutaeus*; after slaughter 24 h–48 h, central temperature 2 ± 2 °C; pH 5.65 ± 0.01; protein 20.37 ± 0.51%) and pork back-fat (89.63 ± 0.81%) were provided by a local slaughterhouse (Xinxiang, China). Removing the visible fat and connective tissue, the pork meat was ground with a 6 mm hole plate using a meat grinder (JR-120; Zhucheng, China) and mixed uniformly. After that, the meat was mixed uniformly and packaged in nylon/PE bags (200 g/bag), and then stored at 2 ± 2 °C until use within 24 h. The KCl, MgCl_2_, K_2_HPO_4_, KH_2_PO_4_, sodium chloride, and sodium bicarbonate (analytically pure) were supplied by Tianjin Boddi Chemical Co., Ltd., Tianjin, China. Tris, EDTA, EGTA, NaN3, DTNB, BSA, and glycine (analytically pure) were purchased from Sigma-Aldrich Co., Darmstadt, Germany.

### 3.2. Extraction of Myofibrillar Protein

The myofibrillar protein was extracted from pork meat as described by Kang et al. [5]. The final myofibrillar protein was acquired and utilized within 48 h. Myofibrillar protein content was measured by biuret using bovine serum albumin (BSA) as the standard.

### 3.3. Preparation of Myofibrillar Protein Solution and Ultrasound Treatment

The pork myofibrillar protein was diluted to 5 mg/mL (dissolved in 50 mmol/L K_2_HPO_4_/KH_2_PO_4_, pH 6.0). The formulas were as follows: the myofibrillar protein at 100 g; T1 contained salt at 2.5 g; sodium bicarbonate at 0 g; T2, T3, and T4 contained salt at 2 g; and sodium bicarbonate at 0.2 g. The myofibrillar protein solutions were homogenized (T25 digital; IKA Ltd., Staufen, Germany) at 2000 rpm for 10 s and left for 30 min to let the solution become uniform. Then, the protein was vacuum-packaged for ultrasonic treatment using an ultrasonic machine (SB-4200, Ningbo Xinzhi Biotechnology Co., Ltd., Ningbo, China) at 2 ± 2 °C. The ultrasound conditions were as follows: the power was 192 W; the frequency was 40 kHz; the ultrasound time of T1 and T2 was 0 min; T3 was 30 min; and T4 was 60 min.

### 3.4. pH

Approximately 20 mL of myofibrillar protein solution (5 mg/mL) was treated using a homogenizer (high-speed homogenizer; Ningbo Xinzhi Biotechnology Co., Ltd., China) at 1500 rpm for 10 s in an ice bath, and the pH was measured using a digital pH meter (PHS-2F; Shanghai Electrical Instrument Co., Ltd., Shanghai, China).

### 3.5. Turbidity

The myofibrillar protein was diluted to 2 mg/mL and measured using an Ultraviolet–visible spectrophotometer (UT-1810; Beijing Purkinje General Instrument Ltd., Beijing, China) with appropriate blank reading absorbance at 320 nm.

### 3.6. Particle Size and Zeta Potential

The measurement of particle size and Zeta potential refers to the method of Zou, Kang, Li, and Ma [45]. To measure the Zeta potential and particle size, the myofibrillar protein solution was diluted to 0.1 mg/mL and then measured with a Zetasizer v7.11 laser nanoparticle sizer (Zetasizer Nano-ZS 90; Malvern, UK).

### 3.7. Active Sulfhydryl

Approximately 1.5 mL of the 5 mg/mL myofibrillar protein solution treated by ultrasound-assisted sodium bicarbonate was suspended in 10 mL of urea-free Tris-glycine buffer (0.086 mol/L Tris, 0.09 mol/L glycine, 4 mmol/L EDTA, pH 8.0). The above samples were vigorously shaken with 50 μL Ellman reagent (4 mg DTNB dissolved in 1 mL of Tris-glycine buffer) in a water bath for 1 h at 25 ± 1 °C. After centrifugation at 12,000× *g* for 10 min, meanwhile, the supernatant without DTNB was taken as the control, and the absorbance was measured at 412 nm (UT-1810; Beijing Purkinje General Instrument Ltd., China). The active sulfhydryl content was calculated according to the formula:Thiol content/(mol/g) = 73.53 × A/ρ(1)

In the equation: 73.53 = 106/1.36104, 1.36104 is the molar extinction coefficient/(cm/mol), and ρ is the mass concentration of the sample protein (mg/mL).

### 3.8. Surface Hydrophobicity

The surface hydrophobicity of protein was determined by the content of the bound hydrophobic chromophore bromophenol blue (BPB) solution and was determined according to the method of Chelh, Gatellier, and Santelhoutellier [46]. The specific operation was as follows: the processed protein solution was adjusted to 2 mg/mL, then 1 mL of 20 mmol (pH = 6) phosphate buffer was added to 200 μL BPB (1 mg/mL) to measure the absorbance. After adjusting the phosphate buffer, the 1 mL of myofibrillar protein solution taken from different test groups was added to a 200 μL BPB oscillation reaction for 10 min and centrifuged for 15 min, and the absorbance of the supernatant was measured using a UV-scanning spectrophotometer at 595 nm after being diluted 10 times. The BPB bound (H_0_) of the protein surface was calculated according to the following formula:BPB bound (μg) = 200 μg × (OD_control_ − OD_sample_)/OD_sample_(2)

### 3.9. Fluorescence Emission Spectra

According to the method of Jin et al. [41], the fluorescence of tryptophan was measured using the fluorescence spectrophotometer (F-180; Tianjin Guangdong Technology Development Co., Ltd., Tianjin, China). The draw of 3.5 mL (0.2 mg/mL) was placed in a quartz Petri dish with a light range of 1 cm. The measurement conditions were as follows: excitation at 283 nm at room temperature, recording the emission spectrum of 300–400 nm, the width of excitation and emission slit was 10 nm, and the data collection rate was 500 nm/min. The change in its internal fluorescence intensity was analyzed.

### 3.10. Statistical Analysis

In the experiment, different added amounts of sodium bicarbonate, salt, and ultrasound time were used at 4 different times (*n* = 4). Test results were expressed as the mean ± standard error (SE). Data were analyzed through the general linear model (GLM) procedure, which considers the treatments (sodium bicarbonate, salt, and ultrasound time) as a fixed effect and the replicates as a random effect. Significant differences between means were identified by the LSD procedure. The difference between means was considered significant at *p* < 0.05.

## 4. Conclusions

This study showed that ultrasound-assisted sodium bicarbonate significantly affected the aggregation and conformation of pork myofibrillar protein. The increase in ultrasound time and the addition of sodium bicarbonate caused the pH to increase significantly, and the active sulfhydryl and hydrophobic groups embedded in the molecules were gradually exposed, leading to a decrease in turbidity and an increase in the active sulfhydryl and surface hydrophobicity. Meanwhile, due to the changes in the tryptophan microenvironment and protein structures, a decrease in the endogenous fluorescence intensity and a significant redshift of the fluorescence peak were observed. Overall, ultrasound-assisted sodium bicarbonate treatment can effectively enhance the solubility of the myofibrillar protein, and may be used to improve the processing quality of reduced-salt meat products.

## Figures and Tables

**Figure 1 molecules-27-07493-f001:**
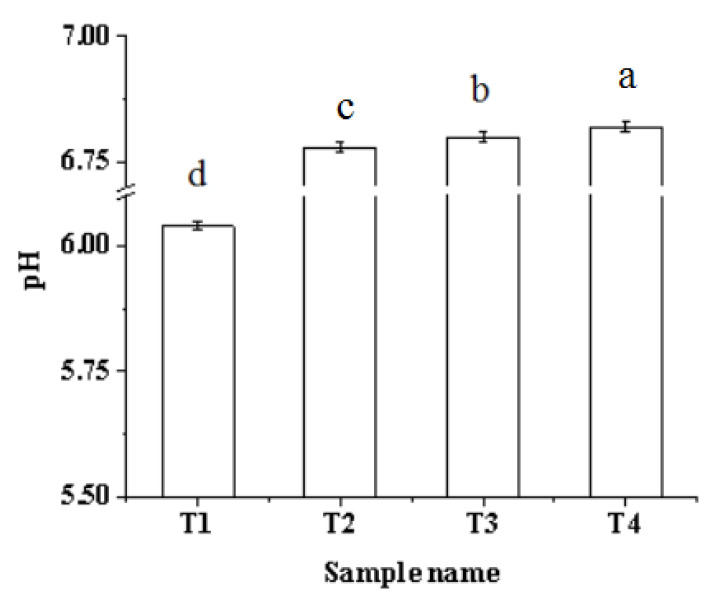
Effects of ultrasound-assisted sodium bicarbonate treatment on the pH of raw pork myofibrillar protein. T1, sodium bicarbonate 0 g, ultrasound time 0 min; T2, sodium bicarbonate 0.2 g, ultrasound time 0 min; T3, sodium bicarbonate 0.2 g, ultrasound time 30 min; T4, sodium bicarbonate 0.2 g, ultrasound time 60 min. Each value represents the mean ± SE, *n* = 4. ^a–d^ Different parameter superscripts indicate significant differences (*p* < 0.05).

**Figure 2 molecules-27-07493-f002:**
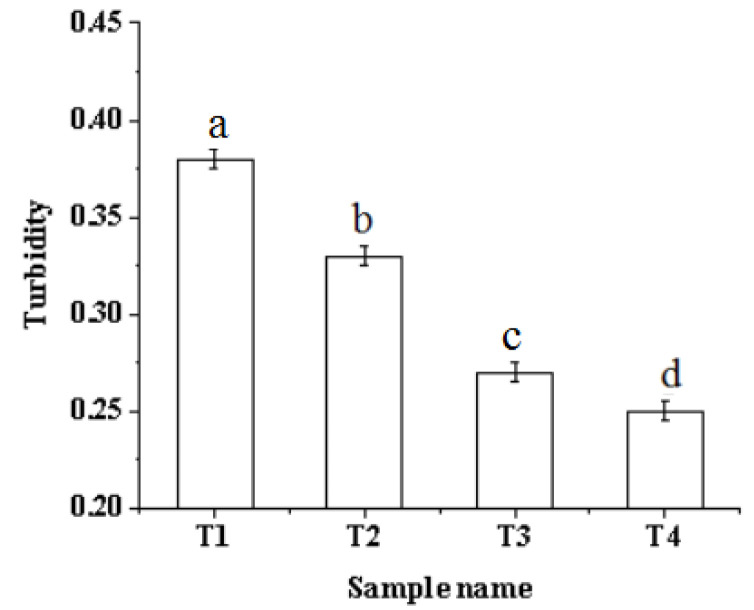
Effects of ultrasound-assisted sodium bicarbonate treatment on the turbidity of raw pork myofibrillar protein. T1, sodium bicarbonate 0 g, ultrasound time 0 min; T2, sodium bicarbonate 0.2 g, ultrasound time 0 min; T3, sodium bicarbonate 0.2 g, ultrasound time 30 min; T4, sodium bicarbonate 0.2 g, ultrasound time 60 min. Each value represents the mean ± SE, *n* = 4. ^a–d^ Different parameter superscripts indicate significant differences (*p* < 0.05).

**Figure 3 molecules-27-07493-f003:**
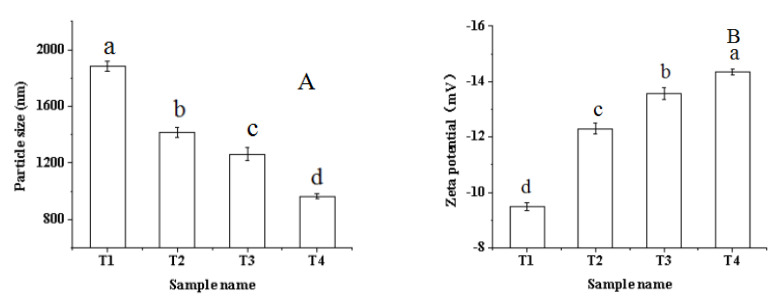
Effects of ultrasound-assisted sodium bicarbonate treatment on the particle size (**A**) and Zeta potential (**B**) of raw pork myofibrillar protein. T1, sodium bicarbonate 0 g, ultrasound time 0 min; T2, sodium bicarbonate 0.2 g, ultrasound time 0 min; T3, sodium bicarbonate 0.2 g, ultrasound time 30 min; T4, sodium bicarbonate 0.2 g, ultrasound time 60 min. Each value represents the mean ± SE, *n* = 4. ^a–d^ Different parameter superscripts indicate significant differences (*p* < 0.05).

**Figure 4 molecules-27-07493-f004:**
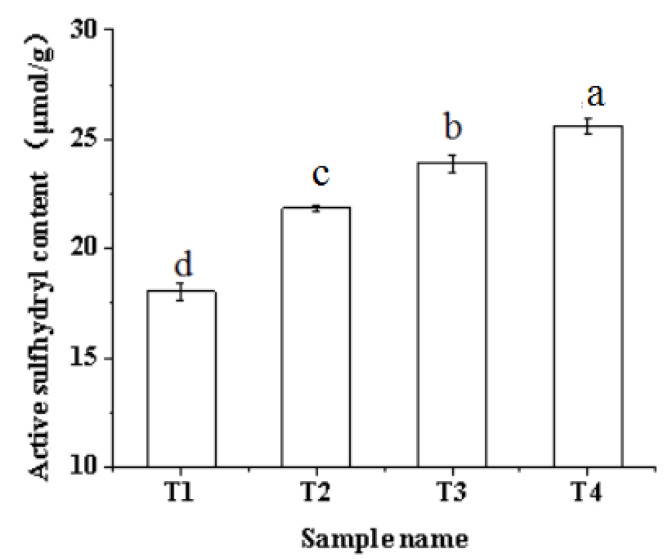
Effects of ultrasound-assisted sodium bicarbonate treatment on the active sulfhydryl of raw pork myofibrillar protein. T1, sodium bicarbonate 0 g, ultrasound time 0 min; T2, sodium bicarbonate 0.2 g, ultrasound time 0 min; T3, sodium bicarbonate 0.2 g, ultrasound time 30 min; T4, sodium bicarbonate 0.2 g, ultrasound time 60 min. Each value represents the mean ± SE, *n* = 4. ^a–d^ Different parameter superscripts indicate significant differences (*p* < 0.05).

**Figure 5 molecules-27-07493-f005:**
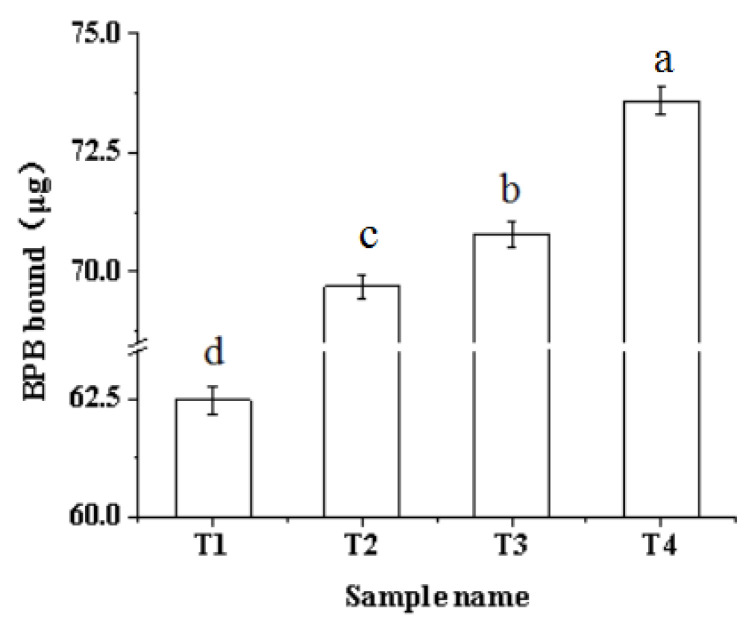
Effects of ultrasound-assisted sodium bicarbonate treatment on the surface hydrophobic of raw pork myofibrillar protein. T1, sodium bicarbonate 0 g, ultrasound time 0 min; T2, sodium bicarbonate 0.2 g, ultrasound time 0 min; T3, sodium bicarbonate 0.2 g, ultrasound time 30 min; T4, sodium bicarbonate 0.2 g, ultrasound time 60 min. Each value represents the mean ± SE, *n* = 4. ^a–d^ Different parameter superscripts indicate significant differences (*p* < 0.05).

**Figure 6 molecules-27-07493-f006:**
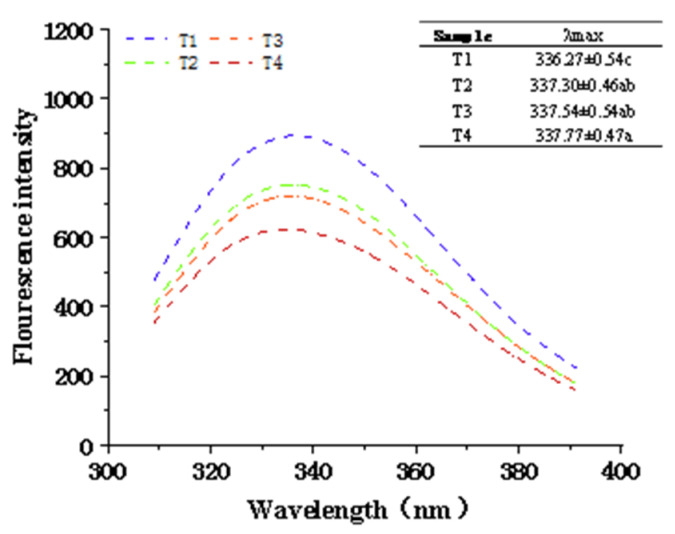
Effects of ultrasound-assisted sodium bicarbonate treatment on the fluorescence emission spectra of raw pork myofibrillar protein. T1, sodium bicarbonate 0 g, ultrasound time 0 min; T2, sodium bicarbonate 0.2 g, ultrasound time 0 min; T3, sodium bicarbonate 0.2 g, ultrasound time 30 min; T4, sodium bicarbonate 0.2 g, ultrasound time 60 min Each value represents the mean ± SE, n = 4. ^a–c^ Different parameter superscripts indicate significant differences (*p* < 0.05).

## Data Availability

Research data are not shared.

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
