# Peer review of "Effect of Ultrasound-Assisted Sodium Bicarbonate Treatment on Aggregation and Conformation of Reduced-Salt Pork Myofibrillar Protein"

_molecules, 2022, doi:10.3390/molecules27217493_

Round 1

Reviewer 1 Report

The manuscript “Effect of ultrasound-assisted sodium bicarbonate treatment on aggregation and conformation of reduced-salt pork myofibrillar protein" is devoted to a study of the aggregation and conformation of pork myofibrillar protein. The effect of exposure to ultrasound and bicarbonate has been studied in detail.  Surface hydrophobicity, turbidity, particle size and zeta potential were measured and discussed. In general, the work may be interesting for specialists in the field of protein chemistry and useful from the point of view of the future organization of industrial processes. I believe that the work is mistakenly attributed to the "Review" type.

The seems somewhat incomplete, although the methods are given quite fully, the data obtained are discussed in sufficient detail, and the necessary conclusions are drawn. I think, this manuscript can be published in the Molecules after major revision taking into account and some of the remarks described below:

1.      Add manufacturers and purity of reagents, please.

2.      The article completely lacks the data obtained in the form of tables and graphs. This greatly complicates the visual perception of the results obtained. The data should be added in the form mentioned above.

Author Response

The manuscript “Effect of ultrasound-assisted sodium bicarbonate treatment on aggregation and conformation of reduced-salt pork myofibrillar protein" is devoted to a study of the aggregation and conformation of pork myofibrillar protein. The effect of exposure to ultrasound and bicarbonate has been studied in detail. Surface hydrophobicity, turbidity, particle size and zeta potential were measured and discussed. In general, the work may be interesting for specialists in the field of protein chemistry and useful from the point of view of the future organization of industrial processes. I believe that the work is mistakenly attributed to the "Review" type.

Response: Thanks for your good suggestion. We have reviewed the mistake in the reviewed manuscript (Line 1), and hope to meet the Journal's requirements.

The seems somewhat incomplete, although the methods are given quite fully, the data obtained are discussed in sufficient detail, and the necessary conclusions are drawn. I think, this manuscript can be published in the Molecules after major revision taking into account and some of the remarks described below:

1. Add manufacturers and purity of reagents, please.

Response: Thanks for your good question. We have added manufacturers and purity of reagents in the reviewed manuscript (Line 86-89).

2. The article completely lacks the data obtained in the form of tables and graphs. This greatly complicates the visual perception of the results obtained. The data should be added in the form mentioned above.

Response: Thanks for your good suggestion. We are sorry for our mistake, we have added the graphs in the reviewed manuscript.

Reviewer 2 Report

The reported work provides useful information on the phenomena at the molecular level, responsible for desirable changes in the texture of battered pork meat. According to the bibliography referred to by the authors, the present results are part of a series of previous works that have been providing information on the effects of the combination of the use of bicarbonate and ultrasound to obtain texture and salt reduction in meat-based pasta. pig

It is suggested to review the syntax of the texts.

It is suggested to check if the term biurerea is correct, probably the correct term is biuret?

Author Response

The reported work provides useful information on the phenomena at the molecular level, responsible for desirable changes in the texture of battered pork meat. According to the bibliography referred to by the authors, the present results are part of a series of previous works that have been providing information on the effects of the combination of the use of bicarbonate and ultrasound to obtain texture and salt reduction in meat-based pasta.

1. It is suggested to review the syntax of the texts.

Response: Thanks for your good suggestion. We have rewritten the paper according to your advise and the requirement of the Journal, such as manuscript, figures, and so on, and hope to meet the Journal's requirements.

2. It is suggested to check if the term biurerea is correct, probably the correct term is biurerea?

Response: Thanks for your good suggestion. We have reviewed the mistake, and have changed “biurerea” into “biuret” in the reviewed manuscript (Line 93).

Reviewer 3 Report

The work is interesting but there are errors in the manuscript and a deeper discussion is necessary. The figures are missing from the manuscript. 

p1 L25-26. myofibrillar proteins?

p2 L48. Use "about" instead of "approximately".

p2 L89. What is "biurerea"? Please explain.

p3 L119. Delete the "of" in Ellman reagent.

p3 L129. The reference for the surface hydrophobicity method is wrong. Kang et al. (2021b) use the ANS fluorescence technique and NOT the bromophenol blue (BPB) method. Please include a correct reference.

p4 L141. Please use "Petri" instead of "cutri".

p4 L156, 175, etc. The figures are not included in the manuscript!

p5 L205-211. The authors indicate that the exposure of negatively charged amino acids is the main reason for a more negative zeta potential in the proteins. This may be true but most reports indicate that at basic values of pH, there is a ionization of the amino acids free carboxyl groups on the surface and therefore the zeta potential becomes more negative.

p6 L241-262. Please explain the increase in solubility with the concomitant increase in surface hydrophobicity. Walayat et al. (2021) found a logical decrease in solubility in myofibrillar proteins when surface hydrophobicity increased. See https://doi.org/10.3390/antiox10081186

Author Response

The work is interesting but there are errors in the manuscript and a deeper discussion is necessary. The figures are missing from the manuscript.

Response: Thanks for your good suggestion. We are sorry for our mistake, we have added the graphs in the reviewed manuscript, and rewritten the paper according to your advise and the requirement of the Journal, such as manuscript, figures, and so on, and hope to meet the Journal's requirements.

p1 L25-26. myofibrillar proteins?

Response: Thanks for your good question. We have changed “myofibrillar” into “myofibrillar proteins” in the reviewed manuscript (Line 25-26).

p2 L48. Use "about" instead of "approximately".

Response: Thanks for your good question. We have added changed “about” into “approximately” in the reviewed manuscript (Line 48).

p2 L89. What is "biurerea"? Please explain.

Response: We are sorry for our mistake. We have changed “biurerea” into “biuret” in the reviewed manuscript (Line 93).

p3 L119. Delete the "of" in Ellman reagent.

Response: Thanks for your good suggestion. We have deleted the "of" in Ellman reagent in the reviewed manuscript (Line 124).

p3 L129. The reference for the surface hydrophobicity method is wrong. Kang et al. (2021b) use the ANS fluorescence technique and NOT the bromophenol blue (BPB) method. Please include a correct reference.

Response: We are sorry for our mistake. We have reviewed the mistake in the reviewed manuscript (Line 131-135) and reference was as follows:

21. Chelh, I., Gatellier, P., & Santelhoutellier, V. (2006). Technical note: A simplified procedure for myofibril hydrophobicity determination. Meat Science, 74(4), 681-683.

p4 L141. Please use "Petri" instead of "cutri".

Response:  We are sorry for our mistake. We have changed “cutri” into “Petri” in the reviewed manuscript (Line 148).

p4 L156, 175, etc. The figures are not included in the manuscript!

Response: Thanks for your good suggestion. We are sorry for our mistake, we have added the graphs in the reviewed manuscript.

p5 L205-211. The authors indicate that the exposure of negatively charged amino acids is the main reason for a more negative zeta potential in the proteins. This may be true but most reports indicate that at basic values of pH, there is a ionization of the amino acids free carboxyl groups on the surface and therefore the zeta potential becomes more negative.

Response: Thanks for your good suggestion. We reviewed the question in the reviewed manuscript (Line 239-243)

p6 L241-262. Please explain the increase in solubility with the concomitant increase in surface hydrophobicity. Walayat et al. (2021) found a logical decrease in solubility in myofibrillar proteins when surface hydrophobicity increased. See https://doi.org/10.3390/antiox10081186

Response: Thanks for your good question. We have explain the question in the reviewed manuscript (Line 310-313) and reference 41.

Round 2

Reviewer 1 Report

I thank the authors for their corrections. I believe that the manuscript can be published in this fo

Reviewer 3 Report

The authors did a good work including all the corrections and the manuscript is adequate for publication.